# On Policy Reuse: An Expressive Language for Representing and Executing General Policies that Call Other Policies

**Primary Keywords:** *(4) Theory; (8) Knowledge Representation/Engineering*

## Abstract

Recently, a simple but powerful language for expressing and learning general policies and problem decompositions (sketches) has been introduced, which is based on collections of rules defined on a set of Boolean and numerical features. In this work, we consider three extensions of this basic language aimed at making policies and sketches more flexible and reusable: internal memory states, as in finite state controllers indexical features, whose values are a function of the state and a number of internal registers that can be loaded with objects; and modules that wrap up policies and sketches and allow them to call each other by passing parameters. In addition, unlike general policies that select actions indirectly by selecting state transitions, the new language allows for the selection of ground actions directly. The expressive power of the resulting language for recombining policies and sketches is illustrated through a number of examples. The problem of learning policies and sketches in the new language, from the bottom up, is left for future work.

## Introduction

A new language for representing problem structure explicitly has been introduced recently in the form of *sketches* (Bonet and Geffner 2023). Sketches are collections of rules of the form $C \mapsto E$ defined over a set of Boolean and numerical domain features $\Phi$, where $C$ expresses Boolean conditions on the features, and $E$ expresses qualitative changes in their values. Each sketch rule captures a subproblem: the problem of going from a state $s$ whose feature values satisfy the condition $C$, to a state $s'$ where the feature values change with respect to $s$ in agreement with $E$. The language of sketches is powerful as it can encode everything from simple goal serializations to full general policies.

The width of a sketch for a class of problems bounds the complexity of solving the resulting subproblems (Bonet and Geffner 2023). For example, a sketch of width $k$ decomposes problems into subproblems that are solved by the IW algorithm in time exponential in $k$ (Lipovetzky and Geffner 2012). Sketches provide a direct generalization of policies, which are sketches of width $0$ that result in subproblems that can be solved in a single step (Bonet and Geffner 2023).

A method for learning sketches is developed by Drexler, Seipp, and Geffner (2022) where both the rules and its features are obtained by solving a combinatorial optimization problem that accepts three inputs: an upper bound on the width of the sketch to be learned, a small number of domain instances, and a pool of features defined from the domain predicates in a domain-independent manner.

The current language of policies and sketches, however, does not support the *reuse* of existing policy sketches, and assumes instead that the top goals are set up externally.[1] Yet for *reusing* policies and sketches, goals must be set up internally as well. E.g., for reusing a general policy $\pi_{on}$ that achieves an atom $on(x, y)$ for any blocks $x$ and $y$ within a policy $\pi_T$ that builds a given tower, it is necessary for the policy $\pi_T$ to invoke policy $\pi_{on}$ while selecting the right sequence of arguments $x$ and $y$ to pass to the policy.

In this work, we develop an extension of the language of policies and sketches that accommodates policy and sketch reuse. In the resulting framework, policies and sketches can call other policies and sketches while passing them suitable arguments. For example, one can learn a policy $\pi_{on}$ for putting one block on top of another, and use the learned policy for learning a policy $\pi_T$ for building a tower of blocks, and reuse this policy in turn for learning to build any configuration of blocks.

The language extensions are basically three: *memory states*, as in finite state controllers for sequencing behaviors, *indexical features*, whose values are a function of the state and a number of internal registers that can be loaded with objects, and *modules*, that package policies and sketches so that they can be reused by other policies or sketches. In this paper, we do not address the *learning* problem, but the *representation* problem, as in order to learn these policies bottom up, one must be able to represent them.

The paper is structured as follows. The next two sections cover related work and review planning and sketches. Then, two sections introduce extended policies and sketches, and their semantics, and the following section introduces modules, call rules, and action calls. The last section contains a discussion with conclusions and directions for future work.

---

[1] For each predicate $p$ appearing in the goal, a new predicate $p_G$ is introduced with the same arity as $p$. The atom $p(c)$ in the state $s$ means that $p(c)$ is true in $s$, while $p_G(c)$ in $s$ means that $p(c)$ must be true in the goal.

## Related Work

**General policies.** The paper builds on a research thread that introduced the notions of width, generalized rule-based policies and sketches of bounded width, as well as methods for learning them (Lipovetzky and Geffner 2012; Bonet and Geffner 2018, 2023; Francès, Bonet, and Geffner 2021; Drexler, Seipp, and Geffner 2022, 2023). The problem of representing and learning general policies has a long history (Khardon 1999; Martín and Geffner 2004; Fern, Yoon, and Givan 2006), and general plans have also been represented in logic (Srivastava, Immerman, and Zilberstein 2011; Illanes and McIlraith 2019), and neural nets (Groshev et al. 2018; Toyer et al. 2018; Bueno et al. 2019; Rivlin, Hazan, and Karpas 2020; Garg, Bajpai, and Mausam 2020; Ståhlberg, Bonet, and Geffner 2023).

**Planning programs and inductive programming.** Planning programs have been proposed as a language for representing and learning general policies that make use of a number of programming language constructs (Aguas, Celorrio, and Jonsson 2016; Segovia-Aguas, Jiménez, and Jonsson 2019, 2021). However, without a rich feature language for talking about goals, states, and their relation, planning programs are basically limited to families of tasks where the goal is fixed; like "putting all blocks on the table", but not for "building a given tower".[2] The problem of program reuse, which is related to the problem of policy reuse, has been considered in program synthesis and inductive programming (Dumancic, Guns, and Cropper 2021; Ellis et al. 2023).

**Deictic representations.** The use of registers to store objects in the extended language of policies and sketches is closely related to the use of indices and visual markers in deictic or indexical representations (Chapman 1989; Agre and Chapman 1990; Ballard et al. 1996). The computational value of such representations, however, has not been clear (Finney et al. 2013). In our setting, indices (registers) make policies (and sketches) parametric and more expressive, as (sub)policies can be reused by setting and resetting the values of registers as needed.

**Hierarchical RL.** Hierarchical structures have been used in RL in the form of options (Sutton, Precup, and Singh 1999), hierarchies of machines (Parr and Russell 1997) and MaxQ hierarchies (Dietterich 2000), and a vast literature has explored methods for learning hierarchical policies (McGovern and Barto 2001; Machado, Bellemare, and Bowling 2017), in some cases, encoding varying goals as part of the state (Kulkarni et al. 2016; Hafner et al. 2022). In almost all cases, however, policies and subpolicies are learned jointly, not bottom up, and the information that is conveyed by parameters, if they exist, is very limited.

## Planning Problems and Sketches

A **planning problem** refers to a classical planning problem $P = \langle D, I \rangle$ where $D$ is the planning domain and $I$ contains information about the instance; namely, the objects in the instance, the initial situation, and the goal. A class $\mathcal{Q}$ of

---

[2]The goal is fixed in a class $\mathcal{Q}$ of planning problems when for any $P$ in $\mathcal{Q}$, the goal $G$ of $P$ is determined by its initial state.

---

**Algorithm 1: SIW$_R$ search given sketch $R$**

1: **Input:** Sketch $R$ over features $\Phi$ that induces relation $\prec_R$
2: **Input:** Planning problem $P$ with initial state $s_0$ on which the features in $\Phi$ are well defined
3: $s \leftarrow s_0$
4: **while** $s$ is not a goal state of $P$ **do**
5:     Run IW search from $s$ to find goal state $s'$ of $P$, or state $s'$ such that $s' \prec_R s$
6:     **if** $s'$ is not found, **return** FAILURE
7:     $s \leftarrow s'$
8: **return** path from $s_0$ to the goal state $s$

Figure 1: SIW$_R$: sketch $R$ used to decompose problem into subproblems, each solved with the IW algorithm.

planning problems is a set of instances over the same domain $D$. A **sketch** for a class of problems $\mathcal{Q}$ is a set of **rules** of the form $C \mapsto E$ based on **features** over the domain $D$ which can be Boolean or numerical, with non-negative integer values (Bonet and Geffner 2023). The condition $C$ is a conjunction of expressions like $p$ and $\neg p$ for Boolean features $p$, and $n > 0$ and $n = 0$ for numerical features $n$, while the effect $E$ is a conjunction of expressions like $p$, $\neg p$, and $p?$, and $n\downarrow$, $n\uparrow$ and $n?$ for Boolean and numerical features $p$ and $n$, respectively. A state pair $(s, s')$ over an instance $P$ in $\mathcal{Q}$ is **compatible** with a rule $r$ if the state $s$ satisfies the condition $C$, and the change of value for the features from $s$ to $s'$ is consistent with $E$. This is written as $s' \prec_r s$ and $s' \prec_R s$ if $R$ is a set of rules that contains $r$.

A sketch $R$ for a class $\mathcal{Q}$ splits the problems $P$ in $\mathcal{Q}$ into **subproblems** $P[s]$ that are like $P$ but with initial state $s$ (where $s$ is a reachable state in $P$), and goal states $s'$ that are either goal states of $P$, or states $s'$ such that $s' \prec_R s$. The algorithm SIW$_R$ shown in Fig 1 uses this problem decomposition for solving problems $P$ in $\mathcal{Q}$ by solving subproblems $P[s]$ via the IW algorithm (Lipovetzky and Geffner 2012). If the sketch has bounded serialized width over $\mathcal{Q}$ and is terminating, SIW$_R$ solves any problem $P$ in $\mathcal{Q}$ in polynomial time (Bonet and Geffner 2023; Srivastava et al. 2011a).

## Serialized Width, Acyclicity, and Termination

The **width** of a planning problem $P$ provides a complexity measure for finding an optimal plan for $P$. If $P$ has $N$ ground atoms and its width is bounded by $k$, written $w(P) \leq k$, an optimal plan for $P$ can be found in $\mathcal{O}(N^{2k-1})$ time and $\mathcal{O}(N^k)$ space by running the algorithm IW($k$), which is a simple breadth-first search where newly generated states are pruned if they do not make a tuple (set) of $k$ atoms or less true for the first time in the search (Lipovetzky and Geffner 2012). If the width of $P$ is bounded but its value is unknown, a plan (not necessarily optimal) can be found by running the algorithm IW with the same complexity bounds. IW calls IW($i$) iteratively with $i = 0, \ldots, k$, until $P$ is solved (Lipovetzky and Geffner 2012). If $P$ has no solution, its width is defined as $w(P) \doteq \infty$, and if it has a plan of length one, as $w(P) \doteq 0$. The width notion extends to classes $\mathcal{Q}$ of problems: $w(\mathcal{Q}) \leq k$ iff $w(P) \leq k$ for each

problem $P$ in $\mathcal{Q}$. If $w(\mathcal{Q}) \leq k$ holds for class $\mathcal{Q}$, then any problem $P$ in $\mathcal{Q}$ can be solved in **polynomial time** as $k$ is independent of the size of $P$.

A sketch $R$ has **serialized width** bounded by $k$ on a class $\mathcal{Q}$, denoted as $w_R(\mathcal{Q}) \leq k$, if for any $P$ in $\mathcal{Q}$, the possible subproblems $P[s]$ have width bounded by $k$. In such a case, every subproblem $P[s]$ that arises when running the SIW$_R$ algorithm on $P$ can be solved in **polynomial time.**

Finally, a sketch $R$ is **acyclic** in $P$ if there is no state sequence $s_0, s_1, \ldots, s_n$ in $P$ such that $s_{i+1} \prec_R s_i$, for $0 \leq i < n$, and $s_n = s_0$. $R$ is acyclic in $\mathcal{Q}$ if it is acyclic in each problem $P$ in $\mathcal{Q}$. The SIEVE algorithm (Srivastava et al. 2011a) can check whether a sketch $R$ is **terminating**, and hence acyclic, by just considering the rules in $R$ and the graph that they define (Bonet and Geffner 2023). For a terminating sketch $R$ with a bounded serialized width over $\mathcal{Q}$, SIW$_R$ is guaranteed to find a solution to any problem $P$ in $\mathcal{Q}$ in **polynomial time** (Bonet and Geffner 2023).

Example 1 shows a policy (sketch of width 0) for achieving the atom $on(x, y)$ for blocks $x$ and $y$ in any Blocksworld instance, and also a sketch of width 2. The policy picks blocks above $x$ and $y$, and put them away until no more such blocks exist, and then stacks $x$ on $y$. However, the blocks are picked in non-specific order; e.g., one above $x$, one above $y$, another above $x$, etc. See below for a different policy that picks the blocks in order.

---

**Example 1: Policy and sketch for the class $\mathcal{Q}_{on}$**

A general policy $\pi_{on}$ for the class $\mathcal{Q}_{on}$ of all Blocksworld problems with atomic goal $on(x, y)$ for two blocks $x$ and $y$ can be defined with the numerical feature $n$ that counts the number of blocks above $x$ and $y$, and the Booleans $On$ that represents whether $x$ is on $y$, $H$ that represents whether a block is being held, and $H_x$ that represents whether block $x$ is being held. The set of features is $\Phi = \{On, H, H_x, n\}$, and the rules are:

$r_0 := \{\neg On, n > 0, \neg H, \neg H_x\} \mapsto \{n\downarrow, H, H_x?\}$
$r_1 := \{\neg On, n > 0, H\} \mapsto \{\neg H, \neg H_x\}$
$r_2 := \{\neg On, n = 0, \neg H, \neg H_x\} \mapsto \{H, H_x\}$
$r_3 := \{\neg On, n = 0, H, H_x\} \mapsto \{On, n\uparrow, \neg H, \neg H_x\}.$

Rule $r_0$ says that in states where $n > 0$ and no block is held, picking a block above $x$ or $y$ is good (are subgoals according to the rules), $r_1$ that in states where $n > 0$ and a block is held, putting the held block away from $x$ or $y$ is good, $r_2$ that in states where $n = 0$ and no block is held, picking up $x$ is good, and $r_3$ that in states where $n = 0$ and block $x$ is held, putting $x$ on $y$ is good, which results in $On$ becoming true.

Alternatively, a sketch of width 2 for $\mathcal{Q}_{on}$ is obtained by replacing rules $r_0$ and $r_1$ with the rule $\{\neg On, n > 0\} \mapsto \{n\downarrow, \neg H\}$, and rules $r_2$ and $r_3$ with $\{\neg On, n = 0\} \mapsto \{On, n\uparrow, \neg H\}$.

---

## Extended Sketches

The first two extensions of sketches are introduced.

### Finite Memory

The first language extension adds memory in the form of a finite number of memory states $m$:

**Definition 1** (Sketches with memory). *A sketch with finite memory is a tuple $\langle M, \Phi, m_0, R \rangle$ where $M$ is a finite set of memory states, $\Phi$ is a set of features, $m_0 \in M$ is the initial memory state, and $R$ is a set of rules extended with memory states. Such rules have the form $(m, C) \mapsto (E, m')$ where $C \mapsto E$ is a standard sketch rule, and $m$ and $m'$ are memory states in $M$.*

When the current memory state is $m$, only rules of the form $(m, C) \mapsto (E, m')$ apply. If the current state and memory are $s$ and $m$ respectively, and $s'$ is a state reachable from $s$ such that the pair $(s, s')$ is compatible with the rule $C \mapsto E$, then moving to state $s'$ and setting the memory to $m'$ is compatible with the extended rule $(m, C) \mapsto (E, m')$. In displayed listings, rules $(m, c) \mapsto (E, m')$ are written as $m \,\|\, \{C\} \mapsto \{E\} \,\|\, m'$ to improve readability.

Example 2 shows a general policy for solving Towers of Hanoi with 3 pegs. The policy uses two memory states to alternate between two types of movements. Such a policy cannot be expressed without memory states.

---

**Example 2: General policy for Hanoi with 3 pegs**

Liu et al. (2023) describe a simple policy for the class $\mathcal{Q}_{Hanoi}$ that consists of Towers-of-Hanoi instances with 3 pegs for moving a tower in the first peg to the third peg. The policy is expressed by referring to the relative size of the disk being moved, among the top disks at each peg, either a movement of the smallest disk, or a movement of the other disk:

*Alternate actions between the smallest disk and a non-smallest disk. When moving the smallest disk, always move it to the left. (If it is in the first peg, move it to the third peg.) If the smallest disk is on the first pillar, move it to the third one. When moving the non-smallest disk, take the only valid action.*

The movements are expressed in the language of sketches with three Boolean features $p_{i,j}$, $1 \leq i < j \leq 3$, that are true if the top disk at peg $i$ is smaller than the top disk at peg $j$. For example, the smallest disk is at peg 1 (resp. peg 3) iff $p_{1,2} \wedge p_{1,3}$ (resp. $\neg p_{1,3} \wedge \neg p_{2,3}$) holds. The alternation of movements is obtained by using two memory states $m_0$ and $m_1$ of which $m_0$ is the initial memory. The rules are:

*% Movements of the smallest disk*
$m_0 \,\|\, \{p_{1,2}, p_{1,3}\} \mapsto \{p_{1,2}?, \neg p_{1,3}, \neg p_{2,3}\} \,\|\, m_1$
$m_0 \,\|\, \{\neg p_{1,2}, p_{2,3}\} \mapsto \{p_{1,2}, p_{1,3}, p_{2,3}?\} \,\|\, m_1$
$m_0 \,\|\, \{\neg p_{1,3}, \neg p_{2,3}\} \mapsto \{\neg p_{1,2}, p_{1,3}?, p_{2,3}\} \,\|\, m_1$

*% Movements of the other disk*
$m_1 \,\|\, \{p_{1,2}, p_{1,3}, p_{2,3}\} \mapsto \{\neg p_{2,3}\} \,\|\, m_0$
$m_1 \,\|\, \{p_{1,2}, p_{1,3}, \neg p_{2,3}\} \mapsto \{p_{2,3}\} \,\|\, m_0$
$m_1 \,\|\, \{\neg p_{1,2}, p_{1,3}, p_{2,3}\} \mapsto \{\neg p_{1,3}\} \,\|\, m_0$
$m_1 \,\|\, \{\neg p_{1,2}, \neg p_{1,3}, p_{2,3}\} \mapsto \{p_{1,3}\} \,\|\, m_0$
$m_1 \,\|\, \{p_{1,2}, \neg p_{1,3}, \neg p_{2,3}\} \mapsto \{\neg p_{1,2}\} \,\|\, m_0$
$m_1 \,\|\, \{\neg p_{1,2}, \neg p_{1,3}, \neg p_{2,3}\} \mapsto \{p_{1,2}\} \,\|\, m_0$

---

## Registers, Indexicals, Concepts, and Roles

The second extension introduces internal memory in the form of *registers*. Registers store objects, which can be referred to in features that become indexical or parametric, as their value changes when the object in the register

changes; e.g., the number of blocks above the block in register zero. The objects that can be placed into the registers $\mathfrak{R} = \{\mathfrak{r}_0, \mathfrak{r}_1, \ldots\}$ are selected by two new classes of features called *concepts* and *roles* that denote sets of objects and set of object pairs, respectively (Baader et al. 2003).

Concept features or simply concepts are denoted in sans-serif font such as 'C', and in a state $s$, they denote the set of objects in the problem $P$ that satisfy the unary predicate 'C' in $s$. Role features or simply roles are also denoted in sans-serif font such as 'R', and in a state $s$, they denote the set of object pairs in the problem $P$ that satisfy the binary relation 'R'. Concept and role features are also used as numerical features, e.g., as conditions 'C $> 0$' or effects 'R$\downarrow$', with the understanding that the corresponding numerical feature is given by the cardinality of the concept C or role R in the state; namely, the number of objects or object pairs in their denotation.

While a plain feature is a function of the problem state, an indexical or parametric feature is a function of the problem state and the value of the registers; like "the distance of the agent to the object stored in $\mathfrak{r}_0$". When the value of a register $\mathfrak{r}$ changes, the denotation of indexical features that depend on the value of the register may change as well. We denote by $\Phi(\mathfrak{r})$ the subset of features in $\Phi$ that refer to (i.e., depend on the value of) register $\mathfrak{r}$. The set of features $\Phi$ is assumed to contain a (parametric) concept for each register $\mathfrak{r}$, whose denotation is the singleton that contains the object in $\mathfrak{r}$, and that is also denoted by $\mathfrak{r}$.

The extended sketch language provides load effects of the form $Load(\mathsf{C}, \mathfrak{r})$ for updating the value or registers for a concept C and register $\mathfrak{r}$; an expression that indicates that the content of the register $\mathfrak{r}$ is to be set to *any object* in the (current) denotation of C, a choice that is *non-deterministic*. Loading an object into register $\mathfrak{r}$ can be thought as placing the *marker* $\mathfrak{r}$ on the object. A rule with effect $Load(\mathsf{C}, \mathfrak{r})$ has the condition C $> 0$ to ensure that C contains some object. Likewise, since a load may change the denotation of features, the effect of a load rule on register $\mathfrak{r}$ is assumed to contain also the extra effects $\phi$? for the features $\phi$ in $\Phi(\mathfrak{r})$, and no other effects. Formally,

**Definition 2** (Extended rules). *An **extended rule** over the features $\Phi$ and memory $M$ has the form $(m, C) \mapsto (E, m')$ where $m$ and $m'$ are memory states, $C$ is a set of conditions of the form $p$, $\neg p$, $n = 0$, and $n > 0$ for Boolean and numerical features $p$ and $n$ in $\Phi$, and $E$ is a set of effects of the form $p$, $\neg p$, $p$? for Boolean $p$, $n\downarrow$, $n\uparrow$, and $n$? for numerical $n$, or it has a single load effect of form $Load(\mathsf{C}, \mathfrak{r})$ for some concept C and register $\mathfrak{r}$. If $E$ contains a $Load(\mathsf{C}, \mathfrak{r})$, it also contains $\phi$? for the features $\phi$ in $\Phi(\mathfrak{r})$, but no other effect.*

Rules with a load effect are called *internal rules*, as they capture changes in the internal memory only, while the other rules are called *external rules*. For simplicity, it is assumed, that each load rule contains a single load effect, and that internal rules have their own memory state, meaning that no external rule can be applied in the same memory where an internal rule is applied. For convenience, we also allow internal rules of the form $(m, C) \mapsto (true, m')$, abbreviated as $(m, C) \mapsto (\{\}, m')$, that permit to change from mem-

ory state $m$ to memory state $m'$ under condition $C$. Memory states associated with internal rules are referred to as *internal memory*; others, as *external memory*. Extended sketches are formally defined as

**Definition 3** (Extended sketch). *An **extended sketch** is a tuple $\langle M, \mathfrak{R}, \Phi, m_0, R \rangle$ where $M$ and $\mathfrak{R}$ are finite sets of memory states and registers, respectively, $\Phi$ is a set of features, $m_0 \in M$ is the initial memory state, and $R$ is a set of extended $\Phi$-rules over memory $M$. An extended sketch is well defined on a class $\mathcal{Q}$ if its set of features $\Phi$ is well defined on $\mathcal{Q}$. If $m$ is a memory state, $R(m)$ denote the subset of rules in $R$ of form $(m, C) \mapsto (E, m')$.*

Example 3 defines a general indexical policy $\pi_{on}^*$ for the class of Blockworld problems $\mathcal{Q}_{on}$ in Example 1. The policy can be understood as putting the *"mark"* $\mathfrak{r}_0$ on a block in N, and a second *mark* $\mathfrak{r}_1$ is moved up from $\mathfrak{r}_0$ to the topmost block above $\mathfrak{r}_0$. The block marked as $\mathfrak{r}_1$ is then put away and the mark $\mathfrak{r}_1$ is updated. When the block in $\mathfrak{r}_0$ becomes clear, the process repeats until counter N becomes zero. When N is zero, both $x$ and $y$ are clear. The policy then picks $x$ and puts it on top of $y$. It is important to notice that this policy guarantees clearing one block $x$ or $y$ after clearing the other while the policy from Example 1 does not have this guarantee. The marks *fix the attention* on the blocks in a common tower which permits the use of much simpler features.

---

**Example 3: Indexical policy for the class $\mathcal{Q}_{on}$**

A general indexical policy $\pi_{on}^*$ for $\mathcal{Q}_{on}$ can be obtained with 8 memory states, two registers $\mathfrak{r}_0$ and $\mathfrak{r}_1$, the concept N for the blocks in $\{x, y\}$ that are not clear, the indexical concept $\mathsf{T}_0$ (resp. $\mathsf{T}_1$) for the block on $\mathfrak{r}_0$ (resp. $\mathfrak{r}_1$), if any, and the Boolean features $A$ that is true if the block in $\mathfrak{r}_1$ is above $x$ or $y$, $H$ (resp. $H_x$) that is true if holding some block (resp. $x$), and $On$ that is true iff $on(x, y)$ holds. The set of features is $\Phi = \{On, H, H_x, A, \mathsf{N}, \mathsf{T}_0, \mathsf{T}_1\}$, the initial memory state is $m_0$, and the rules are:

*% Internal rules*

$r_0 := m_0 \parallel \{\neg H, \mathsf{N} > 0\} \mapsto \{Load(\mathsf{N}, \mathfrak{r}_0), \mathsf{T}_0?\} \parallel m_1$
$r_1 := m_0 \parallel \{\neg H, \mathsf{N} = 0\} \mapsto \{\} \parallel m_6$
$r_2 := m_0 \parallel \{H\} \mapsto \{\} \parallel m_5$
$r_3 := m_1 \parallel \{\mathfrak{r}_0 > 0\} \mapsto \{Load(\mathfrak{r}_0, \mathfrak{r}_1), \mathsf{T}_1?, A?\} \parallel m_2$
$r_4 := m_2 \parallel \{\mathsf{T}_1 > 0\} \mapsto \{Load(\mathsf{T}_1, \mathfrak{r}_1), \mathsf{T}_1?, A?\} \parallel m_2$
$r_5 := m_2 \parallel \{\mathsf{T}_1 = 0\} \mapsto \{\} \parallel m_4$
$r_6 := m_3 \parallel \{\mathsf{T}_0 > 0\} \mapsto \{\} \parallel m_1$
$r_7 := m_3 \parallel \{\mathsf{T}_0 = 0\} \mapsto \{\} \parallel m_0$

*% External rules*

$r_8 := m_4 \parallel \{\neg H, A\} \mapsto \{H, \neg A, \mathsf{N}?\} \parallel m_5$
$r_9 := m_5 \parallel \{H, \neg A\} \mapsto \{\neg H\} \parallel m_3$
$r_{10} := m_5 \parallel \{H, \neg A\} \mapsto \{\neg H, \mathsf{N}\downarrow\} \parallel m_3$
$r_{11} := m_6 \parallel \{\neg H_x\} \mapsto \{H_x\} \parallel m_7$
$r_{12} := m_7 \parallel \{H_x, \neg On\} \mapsto \{On\} \parallel m_7$

where rule $r_0$ puts either block $x$ or $y$ to be cleared in $\mathfrak{r}_0$, $r_1$ goes to $m_6$ when $x$ and $y$ are clear, $r_2$ is used when initially holding a block, $r_3$ puts $\mathfrak{r}_0$ in $\mathfrak{r}_1$, $r_4$ moves $\mathfrak{r}_1$ up in the tower, $r_5$ goes to $m_4$ when $\mathfrak{r}_1$ is clear, rules $r_6, r_7$ form a conditional jump to $m_0$ if $\mathfrak{r}_0$ is clear, and otherwise, jumps to $m_1$, $r_8$ picks block in $\mathfrak{r}_1$, $r_9, r_{10}$ puts it away, $r_{11}, r_{12}$ put block $x$ on $y$. The feature $A$ is used to avoid placing a block being held above $x$ or $y$; this can be seen in rules $r_9$ and $r_{10}$ whose conditions $A$ and $\neg A$ is not mentioned in the effects, meaning that its value

must not change in any transition compatible with the rules.

Indexical features are used to keep track of objects, allowing simpler features than those used in the policy in Example 1. For example, the features $T_0$ and $T_1$, that contain the block directly on $\mathfrak{r}_0$ and $\mathfrak{r}_1$, respectively, are much simpler than the feature $n$ that counts the total number of blocks above $x$ and $y$. The following Figure 2 illustrates the usage of indexicals.

(a) Load($N$,$\mathfrak{r}_0$)  (b) Load($T_1$,$\mathfrak{r}_1$)  (c) Load($T_1$,$\mathfrak{r}_1$)  (d) Load($T_1$,$\mathfrak{r}_1$)

Figure 2: Illustration of finding block $b_3$ that must be put away: (a) the rule $r_0$ sets the marker $\mathfrak{r}_0$ on the block $x \in N$ that must be cleared, (b) the rule $r_3$ sets the marker $\mathfrak{r}_1$ on the block $b_1 \in T_0$ above the block $x$ in $\mathfrak{r}_0$, (c) the rule $r_4$ moves the marker $\mathfrak{r}_1$ one step above to the block $b_2 \in T_1$, (d) the rule $r_4$ moves the marker $\mathfrak{r}_1$ one step above to the topmost block $b_3 \in T_1$. The concept $T_1$ is empty and the loop over $r_4$ terminates.

## Formal Semantics and Termination

Extended sketches are evaluated on planning states augmented with memory states, and values for the registers.

**Definition 4** (Augmented states). *An **augmented state** for a problem $P$ given an extended sketch $\langle M, \mathfrak{R}, \Phi, m_0, R \rangle$ is a tuple $\bar{s} = (s, m, \boldsymbol{v})$ where $s$ is a reachable state in $P$, $m$ is a memory state in $M$, and $\boldsymbol{v}$ is a vector of objects in $Obj(P)^{\mathfrak{R}}$ that tells the content of each register $\mathfrak{r}$, denoted as $\boldsymbol{v}[\mathfrak{r}]$.*

Features $\phi$ are evaluated over pairs $(s, \boldsymbol{v})$ made up of a state $s$ and a value $\boldsymbol{v}$ for the registers. The value for $\phi$ at such a pair is denoted by $\phi(s, \boldsymbol{v})$.

**Definition 5** (Compatible pairs). *Let $r$ be an **external** $\Phi$-rule $(m, C) \mapsto (E, m')$, and let $\boldsymbol{v}$ be a valuation for the registers. A state $s$ satisfies the condition $C$ **given** $\boldsymbol{v}$ if the feature conditions in $C$ are all true in $(s, \boldsymbol{v})$. A state pair $(s, s')$ satisfies the effect $E$ **given** $\boldsymbol{v}$ if the values for the features in $\Phi$ change from $s$ to $s'$ according to $E$; i.e., the following holds where $p$ is a Boolean feature, $n$ is a numerical, concept or role feature, and $\phi$ is any type of feature,*

1. *if $p$ (resp. $\neg p$) is in $E$, $p(s', \boldsymbol{v}) = 1$ (resp. $p(s', \boldsymbol{v}) = 0$),*
2. *if $n{\downarrow}$ (resp. $n{\uparrow}$) is in $E$, $n(s, \boldsymbol{v}) > n(s', \boldsymbol{v})$ (resp. $n(s, \boldsymbol{v}) < n(s', \boldsymbol{v}))$, and*
3. *if $\phi$ is not mentioned in $E$, $\phi(s, \boldsymbol{v}) = \phi(s', \boldsymbol{v})$.*

*The pair $(s, s')$ is **compatible** with an external rule $r = (m, C) \mapsto (E, m')$ **given** $\boldsymbol{v}$, denoted as $s' \prec_{r/\boldsymbol{v}} s$, if given $\boldsymbol{v}$, $s$ satisfies $C$ and the pair satisfies $E$. The pair is compatible with a set of rules $R$ given $\boldsymbol{v}$, denoted as $s' \prec_{R/\boldsymbol{v}} s$, if it is compatible with some external rule in $R$ given $\boldsymbol{v}$.*

The search algorithm for extended sketches, called $\text{SIW}_R^*$ and shown below in Fig. 3, maintains the current memory state and values for registers, implements the internal rules,

and perform IW searches to solve the subproblems that arise when the memory state becomes external.

**Definition 6** (Subproblems). *If $\bar{s} = (s, m, \boldsymbol{v})$ is an augmented state for $P$, where $m$ is external memory, the subproblem $P[\bar{s}]$ is a planning problem like $P$ but with initial state $s$ and goal states that are goal states of $P$ or states $s'$ such that $s' \prec_{r/\boldsymbol{v}} s$ for some rule $r$ in $R(m)$.*

Internal rules do not generate classical subproblems and do not change the planning state $s$ but they affect the memory state and the register values, and with that, the definition of the subproblems that follow. The notion of **reduction** captures how internal rules are processed:

**Definition 7** (Reduction). *Let $\langle M, \mathfrak{R}, \Phi, m_0, R \rangle$ be an extended sketch for a planning problem $P$, and let $(s, m, \boldsymbol{v})$ be an augmented state for $P$ where $m$ is in $M$. The pair $\langle (s, m, \boldsymbol{v}), (s, m', \boldsymbol{v}') \rangle$ is a **reduction step** if there is an internal rule $r$ in $R$ of form $(m, C) \mapsto (E, m')$ such that 1) $s$ satisfies the condition $C$ given $\boldsymbol{v}$, and 2) $\boldsymbol{v}'$ is equal to $\boldsymbol{v}$, except if $\text{Load}(C, \mathfrak{r})$ is in $E$, in which case $\boldsymbol{v}'[\mathfrak{r}] \in C(s, \boldsymbol{v})$. A sequence of reduction steps starting at $(s, m, \boldsymbol{v})$ and ending at $(s, m', \boldsymbol{v}')$ where $m'$ is external is called a **reduction**, and it is denoted by $(s, m, \boldsymbol{v}) \to^* (s, m', \boldsymbol{v}')$. For convenience, if $m$ is external, we also write $(s, m, \boldsymbol{v}) \to^* (s, m, \boldsymbol{v})$.*

An **initial augmented state** for problem $P$ is of the form $(s_0, m, \boldsymbol{v})$ where $s_0$ is the initial state in $P$, and $(s_0, m_0, \boldsymbol{v}_0) \to^* (s_0, m, \boldsymbol{v})$ for the initial memory $m_0$ and some $\boldsymbol{v}_0$ in $Obj(P)^{\mathfrak{R}}$. There may be different initial augmented states for $P$ that differ in their memory and/or the contents of the registers. Each such initial augmented state defines an **initial subproblem** $P[s_0, m, \boldsymbol{v}]$ (cf. Definition 6).

**Definition 8** (Induced subproblems). *Let $\langle M, \mathfrak{R}, \Phi, m_0, R \rangle$ be an extended sketch for a planning problem $P$ with initial state $s_0$. Let us consider a subproblem $P[\bar{s}]$ for $\bar{s} = (s, m, \boldsymbol{v})$ where $m$ is external, and let $s'$ be a state reachable from $s$ with $s' \prec_{r/\boldsymbol{v}} s$ for some rule $r$ in $R(m)$. Then,*

1. *subproblem $P[s', m', \boldsymbol{v}]$ is **induced** by subproblem $P[\bar{s}]$ if $r = (m, C) \mapsto (E, m')$ and $m'$ is external memory.*
2. *Subproblem $P[s', m'', \boldsymbol{v}']$ is **induced** by subproblem $P[\bar{s}]$ if $r = (m, C) \mapsto (E, m')$, $m'$ is internal memory, and $(s', m', \boldsymbol{v}) \to^* (s', m'', \boldsymbol{v}')$.*

*The collection $P^\circ$ of induced subproblems is the smallest set such that 1) $P^\circ$ contains all the initial subproblems, and 2) $P[s', m', \boldsymbol{v}']$ is in $P^\circ$ if $P[s, m, \boldsymbol{v}]$ is in $P^\circ$ and the first subproblem is induced by the second.*

By "jumping" over subproblems $P[s, m, v]$ where $m$ is internal, the definition ensures that the subproblems that make it into $P^\circ$ all have memory states $m$ that are external, and hence represent classical planning problems. The extended sketch $R$ is said to be **reducible** in $P$ if for any reachable augmented state $(s, m, \boldsymbol{v})$ in $P$ where $m$ is an internal state, there is an augmented state $(s, m', \boldsymbol{v}')$ such that $(s, m, \boldsymbol{v}) \to^* (s, m', \boldsymbol{v}')$. A non-reducible sketch is one in which the executions can cycle or get stuck while performing internal memory operations.

**Definition 9** (Sketch width). *The **width** of a reducible sketch $R$ over a planning problem $P$ is bounded by non-negative*

Figure 3: SIW$_R^*$ solves a problem $P$ by using the extended sketch $R$ to decompose $P$ into subproblems that are solved with IW. Completeness of SIW$_R^*$ is captured in Theorem 10.

integer $k$, denoted by $w_R(P) \leq k$, if the width of each subproblem in $P^\circ$ is bounded by $k$. The **width** of a reducible sketch $R$ over a class of problems $\mathcal{Q}$ is bounded by $k$, denoted by $w_R(\mathcal{Q}) \leq k$, if $w_R(P) \leq k$ for each $P$ in $\mathcal{Q}$.

The **serialized width** is zero for the indexical sketch in Example 3, as the sketch represents a policy where each subproblem is solved in a single step.

### Termination for Extended Sketches

Termination is a key property for sketches that guarantees acyclicity, and that only a polynomial number of subproblems may appear on *any* problem $P$ where the features are well defined (Bonet and Geffner 2023). Termination can be tested in polynomial time by a suitable adaptation of the SIEVE algorithm (Srivastava et al. 2011b; Bonet and Geffner 2023). If the extended sketch is reducible, terminating, and has bounded serialized width over a class $\mathcal{Q}$, then any problem $P$ in $\mathcal{Q}$ can be solved in polynomial time with SIW$_R^*$, shown in Fig. 3.

**Theorem 10** (Termination). *If the extended sketch $R$ is* **reducible**, **terminating**, *and has a* **serialized width** *bounded by $k$ over the class of problems $\mathcal{Q}$, then SIW$_R^*$ finds plans for any problem $P$ in $\mathcal{Q}$ in polynomial time.*

### Reusable Modules

Reusable policies and sketches are wrapped into *modules*. A module is a named tuple $\langle args, Z, M, \mathfrak{R}, \Phi, m_0, R \rangle$ where $args = \langle x_1, x_2, \ldots, x_n \rangle$ is a tuple of arguments, each one being either a *static concept or role*, $Z$ and $\Phi$ are sets of features, $M$ is a set of memory states, $\mathfrak{R}$ is a set of registers, $m_0 \in M$ is the initial memory state, and $R$ is a set of rules. The features in $\Phi$ are the ones mentioned in the sketch rules in $R$, and their definition may depend on the value of the

arguments and the features in $Z$. The sketch $R$ in a module can contain two new types of internal rules, *call and do rules,* that permit to call other modules, and to directly execute ground actions of the planning problem, respectively. The value for the arguments in either case are given by the features in $\Phi$ or $Z$, and the arguments of the module; the difference between $\Phi$ and $Z$ is that the rules in $R$ must track the changes for the features in $\Phi$, while $Z$ can be used to define features in $\Phi$, appear in conditions of rules, and provide values to arguments in call and do rules. If the name of the module is name, we refer to it as $\texttt{name}(x_1, x_2, \ldots, x_n)$.

Call and do rules are of the form $(m, C) \mapsto (\texttt{name}(v_1, v_2, \ldots, v_n), m')$ where $m$ is internal memory, $C$ is a condition, name is a module or action schema name, and each value $v_i$ is of an appropriate type: for *do rules*, $v_i$ must be a concept, while for *call rules*, $v_i$ can be either a concept or a role. The idea is that if a *call rule* is used, the sketch associated with the module name is executed until no rules are applicable, and the control is then returned back to the caller at the memory state $m'$. For *do rules*, an applicable ground action of the form $\texttt{name}(o_1, o_2, \ldots, o_n)$, where the object $o_i$ belongs to the denotation of concept $v_i$, must be applied at the current state to make a transition to a successor state, and control is then returned back to the caller at the memory state $m'$.

The execution model for handling modules involves a stack as described below. Modules call each other by passing arguments but do not get back any values. The "side effects" of a module are in the problem state $s$ that must be driven eventually to a goal state.

Example 4 shows the module $\texttt{on}(\mathsf{X}, \mathsf{Y})$ for a policy for the class $\mathcal{Q}_{on}$ that essentially implements the policy in Example 3. However, by directly executing ground actions, we define a policy that uses even simpler features than the ones used in Example 3. That is, the indexical Boolean $A$ is not needed because the block being held is put away on the table with a Putdown action.

*% Internal rules*
$r_0 := m_0 \,\|\, \{\neg H, \mathsf{N} > 0\} \mapsto \{Load(\mathsf{N}, \mathfrak{r}_0), \mathsf{T}_0?\} \,\|\, m_1$
$r_1 := m_0 \,\|\, \{\neg H, \mathsf{N} = 0\} \mapsto \{\} \,\|\, m_6$
$r_2 := m_0 \,\|\, \{H\} \mapsto \{\} \,\|\, m_5$
$r_3 := m_1 \,\|\, \{\mathfrak{r}_0 > 0\} \mapsto \{Load(\mathfrak{r}_0, \mathfrak{r}_1), \mathsf{T}_1?\} \,\|\, m_2$
$r_4 := m_2 \,\|\, \{\mathsf{T}_1 > 0\} \mapsto \{Load(\mathsf{T}_1, \mathfrak{r}_1), \mathsf{T}_1?\} \,\|\, m_2$
$r_5 := m_2 \,\|\, \{\mathsf{T} = 0\} \mapsto \{\} \,\|\, m_4$
$r_6 := m_3 \,\|\, \{\mathsf{T}_0 > 0\} \mapsto \{\} \,\|\, m_1$
$r_7 := m_3 \,\|\, \{\mathsf{T}_0 = 0\} \mapsto \{\} \,\|\, m_0$

*% External rules*
$r_8 := m_4 \,\|\, \{\} \mapsto \texttt{Unstack}(\mathfrak{r}_1, \mathsf{B}) \,\|\, m_5$

$r_9 := m_5 \,\|\, \{\} \mapsto \mathtt{Putdown}(\mathfrak{r}_1) \,\|\, m_3$
$r_{10} := m_6 \,\|\, \{T_x\} \mapsto \mathtt{Pickup}(\mathsf{X}) \,\|\, m_7$
$r_{11} := m_6 \,\|\, \{\neg T_x\} \mapsto \mathtt{Unstack}(\mathsf{X},\mathsf{B}) \,\|\, m_7$
$r_{12} := m_7 \,\|\, \{\} \mapsto \mathtt{Stack}(\mathsf{X},\mathsf{Y}) \,\|\, m_7$

Rules $r_0$–$r_7$ are like the ones for the indexical policy $\pi_{on}^*$. The external rules, however, are do-rules that apply ground actions to remove blocks above $\mathfrak{r}_0$, and to pick $\mathsf{X}$ and put it on $\mathsf{Y}$. The Boolean $T_x$ is used to decide whether to use a $\mathtt{Pickup}$ or $\mathtt{Unstack}$ action to grab $\mathsf{X}$.

Example 5 shows the module $\mathtt{tower}(\mathsf{O},\mathsf{X})$ for building a given tower of blocks, expressed by the object pairs in $\mathsf{O}$ to put in place on top of the block $\mathsf{X}$. The indexical features in the policy, namely $\mathsf{M}$ and $\mathsf{W}$, are conceptually simple. The module calls the modules $\mathtt{on\text{-}table}(\mathsf{X})$ and $\mathtt{on}(\mathsf{X},\mathsf{Y})$, and also calls itself making it a **recursive** module. The module $\mathtt{on\text{-}table}(\mathsf{X})$, that puts the block in the singleton $\mathsf{X}$ on the table, is not spelled out.

---

**Example 5: Module** $\mathtt{tower}(\mathsf{O},\mathsf{X})$

The module $\mathtt{tower}(\mathsf{O},\mathsf{X})$ is aimed at the class $\mathcal{Q}_{tower}$ of problems where blocks are to be stacked in a *single tower* on the table. The goal is described by a conjunction of atoms $\wedge_{i=1}^{k} on(x_i, x_{i-1})$ and $ontable(x_0)$. The module is the tuple $\langle \langle \mathsf{O},\mathsf{X}\rangle, Z, M, \mathfrak{R}, \Phi, m_0, R\rangle$ where $\mathsf{O}$ is a role argument whose denotation contains the pairs $\{(x_i, x_{i-1}) \mid i = 1, ..., k\}$, and $\mathsf{X}$ is a concept argument that denotes the lowest block in the target tower that is misplaced.[a] The other elements in the module are $Z = \emptyset$, $M = \{m_0, m_1, \ldots, m_3\}$, $\mathfrak{R} = \{\mathfrak{r}_0\}$, and a set of features $\Phi = \{\mathsf{M},\mathsf{W}\}$ where $\mathsf{M}$ is the indexical concept that contains the block to be placed above the block in $\mathfrak{r}_0$ according to $\mathsf{O}$, if any, and, $\mathsf{W}$ is the indexical concept that contains the block directly below the block in $\mathfrak{r}_0$, if any, also according to the target tower $\mathsf{O}$. The rules in $R$ are:

*% Module* $\mathtt{tower}(\mathsf{O},\mathsf{X})$
$r_0 := m_0 \,\|\, \{\mathsf{X} > 0\} \mapsto \{Load(\mathsf{X},\mathfrak{r}_0), \mathsf{M}?, \mathsf{W}?\} \,\|\, m_1$
$r_1 := m_1 \,\|\, \{\mathsf{W} = 0\} \mapsto \mathtt{on\text{-}table}(\mathfrak{r}_0) \,\|\, m_2$
$r_2 := m_1 \,\|\, \{\mathsf{W} > 0\} \mapsto \mathtt{on}(\mathfrak{r}_0, \mathsf{W}) \,\|\, m_2$
$r_3 := m_2 \,\|\, \{\mathsf{M} > 0\} \mapsto \mathtt{tower}(\mathsf{O},\mathsf{M}) \,\|\, m_3$

where $r_0$ puts the lowest misplaced block in $\mathfrak{r}_0$, $r_1$ calls the module $\mathtt{on\text{-}table}(\mathfrak{r}_0)$ to place $\mathfrak{r}_0$ on the table, $r_2$ calls $\mathtt{on}$ to place $\mathfrak{r}_0$ on $\mathsf{W}$, and $r_3$ recursively calls with the block that is supposed to be on $\mathfrak{r}_0$.

---
[a] Block $x$ is *well-placed* in state $s$ iff $x$ is on $y$ if the pair $(x,y)$ is in $\mathsf{O}$, and recursively, $y$ is well-placed; it is *misplaced* iff it is not well-placed. In the example, it is also assumed that the lowest block of the target tower must be placed on the table.

---

Finally, Example 6 shows the module $\mathtt{blocks}(\mathsf{O})$ that solves arbitrary instances of Blocksworld. It works by calling the module $\mathtt{tower}(\mathsf{O},\mathsf{X})$ with parameter $\mathsf{X}$ being the singleton that contains the lowest misplaced block in one of the current towers. Such a block is chosen from the concept $\mathsf{L}$ that is a concept of higher complexity. A more involved implementation is able to remove the dependency on $\mathsf{L}$ by first putting a mark on a block $x$ such that $on_G(x,z)$ differs from the atom $on(x,y)$ that is true at the current state, and then moving such feature down with a loop.

---

**Example 6: Module** $\mathtt{blocks}(\mathsf{O})$ **for arbitrary towers**

The module $\mathtt{blocks}(\mathsf{O})$ is aimed at the class $\mathcal{Q}_{blocks}$ of problems for building many target towers. The module takes a single role argument $\mathsf{O}$ whose denotation encodes the pairs $(x,y)$ corresponding to the target $on(x,y)$ atoms as in Example 5. The module is the tuple $\langle \langle \mathsf{O}\rangle, Z, M, \mathfrak{R}, \Phi, m_0, R\rangle$ where $Z = \emptyset$, $M = \{m_0, m_1\}$, $\mathfrak{R} = \{\mathfrak{r}_0\}$, and $\Phi = \{\mathsf{L}\}$ where $\mathsf{L}$ is the concept that contains the lowest misplaced blocks in $\mathsf{O}$.

*% Module* $\mathtt{blocks}(\mathsf{O})$
$r_0 := m_0 \,\|\, \{\mathsf{L} > 0\} \mapsto \{Load(\mathsf{L},\mathfrak{r}_0)\} \,\|\, m_1$
$r_1 := m_1 \,\|\, \{\} \mapsto \mathtt{tower}(\mathsf{O},\mathfrak{r}_0) \,\|\, m_0$

where $r_0$ loads a lowest misplaced block into $\mathfrak{r}_0$, and $r_1$ builds tower starting with the block in $\mathfrak{r}_0$.

---

**Execution Model: SIW$_M$**

The execution model for modules is captured by the SIW$_M$ algorithm in Fig. 4 which uses a stack and a caller/callee protocol, as it is standard in programming languages. It assumes a collection $\{\mathtt{mod}_0, \mathtt{mod}_1, \ldots, \mathtt{mod}_N\}$ of modules where the "entry" module $\mathtt{mod}_0$ is assumed to take no arguments. The execution may involve solving classical planning subproblems, internal operations on the registers, calls to other modules, or execution of ground actions. The modules do not share memory states nor registers, but they may all act on the planning states $s$.

At each time point during execution, there is a single active module $\mathtt{mod}_\ell$ that defines the current set of rules, and there is a current augmented state $(s, m, \boldsymbol{v})$. While no call or do rule is selected, SIW$_M$ behaves exactly as SIW$_R^*$. However, if a call rule $(m, C) \mapsto (\mathtt{mod}_j(x_1, x_2, \ldots, x_n), m')$ is chosen, where $\mathtt{mod}_j$ refers to $\langle \text{args}, Z, M, \mathfrak{R}, \Phi, m_0, R, \rangle$, the following steps are done:

1. Push context $(\ell, \boldsymbol{v}, m')$ where $\boldsymbol{v}$ is value for registers,
2. Set value of arguments of $\mathtt{mod}_j$ to those given by $x_i$,
3. Set memory state to $m_0$ (the initial state of $\mathtt{mod}_j$),
4. Set the current set of rules to $R$,
5. [Cont. execution of $\mathtt{mod}_j$ until no rule is applicable], and
6. Pop context $(\ell, \boldsymbol{v}, m')$, set value of registers to $\boldsymbol{v}$, memory state to $m'$, and rules $R$ to those in $\mathtt{mod}_\ell$.

Similarly, if do rule $(m, C) \mapsto (\text{name}(x_1, x_2, \ldots, x_n), m')$ is chosen, an applicable ground action $\text{name}(o_1, o_2, \ldots, o_n)$ at the current state $s$ with the object $o_i$ in $x_i$, $1 \le i \le n$, is applied and the memory state is set to $m'$. If no such ground action exists, an error code is returned. The SIW$_M$ interpreter has been implemented and all extended, indexical policies and sketches have been tested. We will make the code and the examples available.

## Discussion

A basic, concrete question about policy reuse in a planning setting is: Can a policy for putting one block on top of another be used for building any given tower of blocks, and eventually any block configuration? The question is relevant because it tells us that policies for building given towers and block configurations do not have to be learned from scratch,

**Algorithm 4: Execution model SIW$_M$ for modules**

1: **Input:** Collection $\mathcal{M} = \{\text{mod}_0, \text{mod}_1, \ldots, \text{mod}_N\}$ of modules with entry module $\text{mod}_0$
2: **Input:** Planning problem $P$ with initial state $s_0$ on which the features in $\Phi$ are well defined
3: Initialize stack
4: Let $\mathfrak{R}^j$, $m_0^j$, and $R^j$ be the set of registers, initial memory, and rules of $\text{mod}_j$, $j = 0, 1, \ldots, N$
5: $\ell \leftarrow 0$, $R \leftarrow R^\ell$, and $\bar{s} \leftarrow (s_0, m_0^0, \boldsymbol{v})$ for $\boldsymbol{v} \in \text{Obj}(P)^{\mathfrak{R}^\ell}$
6: **while** $s$ in $\bar{s} = (s, m, \boldsymbol{v})$ is not a goal state of $P$
7:   **if** $m$ is internal memory
8:     Find rule $r = (m, C) \mapsto (E, m')$ with $s, \boldsymbol{v} \vDash C$
9:     **if** $r$ is not found
10:       **if** stack is empty, **return** FAILURE     % Stalled
11:       Pop context $(j, \boldsymbol{v}', m')$ from stack
12:       $\ell \leftarrow j$, $R \leftarrow R^\ell$, $m \leftarrow m'$ and $\boldsymbol{v} \leftarrow \boldsymbol{v}'$
13:     **else**
14:       **if** $Load(\mathsf{C}, \mathfrak{r})$ in $E$, $\boldsymbol{v}[\mathfrak{r}] \leftarrow o$ for some $o \in \mathsf{C}(s, \boldsymbol{v})$
15:       $\bar{s} \leftarrow (s, m', \boldsymbol{v})$
16:   **else**               % $m$ is external memory
17:     Find call/do rule $r = (m, C) \mapsto (E, m')$ with $s, \boldsymbol{v} \vDash C$
18:     **if** $r = (m, C) \mapsto (\text{mod}_j(x_1, x_2 \ldots, x_n), m')$
19:       Push context $(\ell, \boldsymbol{v}, m')$ into stack
20:       $R \leftarrow R^j$, $m \leftarrow m_0^j$     % Hand control to $\text{mod}_j$
21:     **elsif** $r = (m, C) \mapsto (\text{name}(x_1, x_2 \ldots, x_n), m')$
22:       Find ground action $a = \text{name}(o_1, o_2, \ldots, o_n)$ applicable at $s$ with $o_i \in x_i(s, \boldsymbol{v})$, $i = 1, 2, \ldots, n$
23:       **if** there is no such action, **return** FAILURE
24:       $\bar{s} \leftarrow (s', m', \boldsymbol{v})$ where $(s, a, s')$ is transition in $P$
25:     **else**           % No such rule is found
26:       Run IW search from $s$ to find goal state $s'$ of $P$, or
27:         state $s'$ such that $s' \prec_{r/\boldsymbol{v}} s$ for some (external)
28:         rule $r = (m, C) \mapsto (E, m')$ in $R$
29:       **if** no such state is found, **return** FAILURE
30:       $\bar{s} \leftarrow (s', m', \boldsymbol{v})$
31: **return** path from $s_0$ to the goal state $s$

Figure 4: SIW$_M$ uses set of modules $M$ (extended sketches) for solving a problem $P$ via possibly nested calls, execution of ground actions, and IW searches.

but that they can be learned bottom up, from simpler to complex, one after the other (Ellis et al. 2023). The subtlety is that in order for complex policies to use simpler policies, the former must pass the right parameters to the latter depending on the context defined by the top goal and the current state. In this work, we have developed a language for representing policies and sketches, and this form of hierarchical composition.

We have also addressed another source of complexity when learning general policies and sketches: the complexity of the features involved. We have shown that indexical features whose values depend on the content of the registers, that can be dynamically loaded with objects, can be used to drastically reduce the complexity of the features needed, in line with the intuition of the so-called deictic representations, where a constant number of "visual marks" are used to mark objects so that they can be easily referred to (Chapman 1989; Ballard et al. 1996; Finney et al. 2013). In our setting,

an object can be regarded as marked with $\mathfrak{r}$ when the object is loaded into register $\mathfrak{r}$.

The use of registers and concept features allows for general policies that map states into ground actions, even if the set of ground actions change from instance to instance. This is different than general policies defined as filters on state transitions (Bonet and Geffner 2018; Francès, Bonet, and Geffner 2021), which require a model for determining the possible transitions from a state. Policies that map states into actions are more conventional and can be applied model-free.

The achieved expressivity is the result of three extensions in the common language of policies and sketches: internal memory states, like in finite state controllers for sequencing behaviors, indexical concepts and features, whose denotation depends on the value of registers that can be updated, and modules that wrap up policies and sketches and allow them to call each other by passing parameters as a function of the state, the registers, and the goals (represented in the state).

The language of extended policies and sketches adds an interface for calling policies and sketches from other policies and sketches, even recursively, as illustrated in the examples. The resulting language has elements in common with programming languages and planning programs (Aguas, Celorrio, and Jonsson 2016; Segovia-Aguas, Jiménez, and Jonsson 2019, 2021), but there are key differences too. In particular, the use of a rich feature language does not limit policies and sketches to deal with classes of problems with a fixed goal and the policy and sketch modules do not have to represent full procedures or policies; they can also represent sketches where "holes" are filled in with a polynomial IW search when the sketches have bounded width.

Provided with this richer language for policies and sketches, the next step is learning them from small problem instances, adapting the methods developed for standard sketches (Drexler, Seipp, and Geffner 2022). We want to learn hierarchical policies bottom-up by generating and reusing policies, instead of learning them top-down (Drexler, Seipp, and Geffner 2023).

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
