# OpenReview forum: "On Policy Reuse: An Expressive Language for Representing and Executing General Policies that Call Other Policies"
_icaps-conference.org/ICAPS/2024/Conference — ICAPS 2024_

### Official Review · Reviewer_B5hP · 2024-01-12

**Significance And Importance:** 2
**Soundness:** 4
**Novelty:** 3
**Clarity:** 3
**Overall Evaluation:** 1
**Confidence:** 3

**Weaknesses:**

2: No major or minor weaknesses.

**Contributions Of The Paper:**

The paper proposes an extension to an existing language for representing general policies based on rule sketches (Bonet and Geffner 2023).
The extensions are
(1) internal memory states (similar to locations in a finite state automaton);
(2) indexical features based on registers;
and (3) modules that enable to reuse one (simpler) policy, by another (more complex) and introduce rules to directly execute ground actions.

**Ethical Considerations:**

(1) Not Applicable: The paper does not have any ethical considerations to address

**Nomination For Best Paper:**

No

**Questions For Authors:**

In many contexts policies are part of the problem structure to be analyzed, e.g, "Does the policy reach the goal when executed from start?".
The SIW_R algorithm uses (problem decomposition based on) sketch R -- a policy representation -- to solve P.
For a reader less familiar with your line of research, it might be helpful to explicitly state, that, here, the set of possible goal paths is not restricted to be compatible with R, but R guides the SIW search.

Typos:
"internal memory states, as in finite state controllers [,] indexical features" (Abstract)

**Reproducibility:**

4: Authors promise to release code and domains (whichever apply).

**Strengths Of The Paper:**

The contributed extensions enable a hierarchical form of policy composition -- "bottom up, not top-down", which seems particularly useful.
One can represent general policies that are, in principle, not limited to classes with a fixed goal and that can select ground actions of the given problem instance.
The introduction of registers may help to simplify the features needed to represent a policy.

**Weaknesses Of The Paper:**

The paper is of pure theoretical nature in that it only addresses the "representation problem".
The "learning problem" remains open.
Ultimately, this raises the question whether the proposed contribution will be of any practical usage.

---

### Official Review · Reviewer_dakV · 2024-01-22

**Significance And Importance:** 3
**Soundness:** 3
**Novelty:** 3
**Clarity:** 4
**Overall Evaluation:** 2
**Confidence:** 3

**Weaknesses:**

2: No major or minor weaknesses.

**Contributions Of The Paper:**

This paper extends the language for expressing and learning general policies and sketches introduced by Bonet and Geffner. Specifically, it defines internal memory states, indexical features (using registers) and modules to package policies and sketches to allow reuse.

The paper focuses on the representation of these new language features and leaves the learning issue for future work.

**Ethical Considerations:**

(1) Not Applicable: The paper does not have any ethical considerations to address

**Nomination For Best Paper:**

No

**Questions For Authors:**

I understand that the definition of this kind of language allows an agent to learn basic procedures and generalize the resolution of more complex tasks. However, what do you think about how the use of these modules may affect the optimality of the obtained plans?

Have you implemented the algorithms that you introduced in the paper? If so, can you give an example of how a planning problem can be solved?

**Reproducibility:**

0: N/A - nothing to reproduce.

**Strengths Of The Paper:**

The paper is well-organized; it provides sufficient details about the previous work it builds upon to make the paper easy to follow.

Moreover, all the definitions are supported with excellent and incremental examples that help understand what is being defined.

**Weaknesses Of The Paper:**

Maybe describing a brief case study that shows how the algorithms work would result in a more robust version of the paper.

---

### Official Review · Reviewer_PUrG · 2024-01-23

**Significance And Importance:** 3
**Soundness:** 3
**Novelty:** 3
**Clarity:** 3
**Overall Evaluation:** 1
**Confidence:** 5

**Weaknesses:**

0: Minor weaknesses requiring some work to be addressed for the paper to be accepted.

**Contributions Of The Paper:**

The paper outlines an extension to the concept of sketches introduced by Bonet and Geffner (2021, AAAI), extended in 10.48550/arXiv.2012.08033 and further elaborated in 10.48550/arXiv.2311.05490, the latter of which is cited as (Bonet and Geffner, 2023) in this paper.  The extension supports policy reuse and involves adding internal memory states, indexical features, and modules.  These extensions are addressed in the paper along with the introduction of an execution model SIW_M that builds on the modules extension.

**Ethical Considerations:**

(1) Not Applicable: The paper does not have any ethical considerations to address

**Nomination For Best Paper:**

No

**Questions For Authors:**

None

**Reproducibility:**

0: N/A - nothing to reproduce.

**Strengths Of The Paper:**

The paper is clearly an advance in the area and is, in some sense, a natural extension of a line of work going back several years now.  I believe this new way of looking at hierarchies is novel and I believe this paper represents a solid contribution in that line of work.

**Weaknesses Of The Paper:**

I really only have two very minor, and fixable, complaints of the paper:

First, please list algorithms using the standard AAAI form (using horizontal lines, similar to a listing and not as a subfigure within a box).  I thought the use of boxes for the _examples_ was really eye-catching and novel.  But the use of the same kind of boxes for algorithms is very confusing.  I guess I'm so accustomed to seeing algorithms presented in AAAI format as their own blocks, separate from figures and tables, that this one threw me.  And to have those boxes listed inside a figure seemed more odd to me.

Second, I found the citation of a yet unpublished 2023 arxiv paper as the first citation to weaken this paper.  There are several new results in that paper, it's 40 pages long, and ... it's on arxiv instead of a journal or conference.  Yes, I get the message that this is probably under submission to a journal, and, yes, I understand that some of those results are central to this paper.  If I could suggest a slightly different approach: perhaps you could cite the AAAI 2021 paper in the main body and add a footnote that states there are two _substantial_ extensions to the original work (using the arxiv references mentioned above).  Or maybe you could cite all three as independent references.  Either way, this would add two more references to your paper and give a fuller picture of the trajectory of this really interesting line of work.

(This almost a non-issue point, but I have always found it hard to read s' \prec s.  It just looks so wrong to me even after I read the descriptions from the original paper and the extensions.)

---

### Meta-Review · Area_Chair_jdFK · 2024-02-01

**Recommendation:** Accept (Poster)
**Confidence:** 4

**Metareview:**

Clear case. Congrats!

I vote acceptance as poster here as the contribution seems rather technical and specific to a small subset of the community,

**Ethical Considerations:**

(4) Good: The paper adequately addresses most, but not all, of the applicable ethical considerations